# Early Activation of a Multilevel Prostate Cancer Screening Model: Pilot Phase Results and Strategic Perspectives in Lombardy Region

**DOI:** 10.3390/healthcare13162041

**Published:** 2025-08-18

**Authors:** Elena Azzolini, Danilo Cereda, Sara Piccinelli, Michela Viscardi, Silvia Deandrea

**Affiliations:** 1Department of Biomedical Sciences, Humanitas University, Via Rita Levi Montalcini 4, 20072 Milan, Italy; 2IRCCS Humanitas Research Hospital, via Manzoni 56, 20089 Milan, Italy; 3Direzione Generale Welfare, UO Prevenzione, Piazza Città di Lombardia 1, 20124 Milan, Italy; 4Prevention Department, Agency for Health Protection, Viale Indipendenza 3, 27100 Pavia, Italy

**Keywords:** prostate cancer, screening, PSA, population-based program, Lombardy region, screening uptake, early detection, public health policy

## Abstract

**Background/Objectives**: Prostate cancer is the most frequently diagnosed cancer among men in Italy, yet no national population-based screening program exists. In response to new European Council recommendations, the Lombardy Region launched a pilot in November 2024 to assess the feasibility of a digitally enabled, risk-adapted screening model. **Methods**: Men turning 50 were invited to voluntarily self-enroll through the regional electronic health record (FSE). A digital questionnaire assessed eligibility and family history (FH); eligible individuals (97,849 men without a PSA test in the past two years in the regional database) were offered free PSA testing. Risk stratification guided follow-up: men with PSA >3 ng/mL or a positive FH were referred to urology; others were assigned 2- or 5-year recall based on PSA level. **Results**: By June 2025, 8558 men had enrolled (8.7% uptake), 6072 were eligible; 644 (10.6%) reported a positive FH. Among those tested, 58.4% had PSA < 1 ng/mL and were FH-negative, 25.8% had PSA > 1 and <3 ng/mL and were FH-negative, and 15.9% met referral criteria. Digital triage was efficient and ensured care continuity without burdening specialists. Participation varied by local health authority (ATS), depending on implementation context. Preliminary data show a 25.9% reduction in urology consultations and a 35% reduction in MRI use compared to standard care, with no biopsy rate increase. **Conclusions**: The pilot demonstrates the technical feasibility, safety, low administrative burden, and potential sustainability of digital, risk-stratified prostate cancer screening. While participation was low without active invitations, early results support expansion with improved outreach. Lombardy’s experience offers a scalable, EU-aligned model for broader implementation across Italy and other health systems seeking to balance early detection with resource efficiency.

## 1. Introduction

Prostate cancer is the most commonly diagnosed malignancy in men in Europe and a leading cause of cancer death [1]. The potential for early detection through prostate-specific antigen (PSA) testing has been recognized since the 1990s, but population-based prostate cancer screening has long been controversial due to concerns about overdiagnosis and overtreatment of indolent disease [2]. Major randomized trials have demonstrated that PSA-based screening can significantly reduce prostate cancer mortality—for example, the European Randomized Study of Screening for Prostate Cancer (ERSPC) showed a ~20–30% relative mortality reduction at long-term follow-up [3]—but at the cost of a high rate of unnecessary biopsies and detection of low-risk tumors. In a Swedish trial incorporating MRI (the GÖTEBORG-2 study), a screening algorithm that used MRI and targeted biopsies halved the diagnosis of insignificant cancers while maintaining detection of clinically significant cancers [4]. These findings underscore the need for risk-stratified approaches that maximize benefit and minimize harm.

Reflecting the evolving evidence, professional guidelines have shifted in favor of organized, risk-adapted screening. The European Association of Urology (EAU) 2024 guidelines recommend offering well-informed men a risk-tailored early detection strategy, generally starting at age 50 (earlier for higher-risk groups), with interval determination based on baseline PSA level [5]. Men with PSA above ~1 ng/mL at baseline are considered “at risk” and should undergo more frequent follow-up (e.g., every 2 years), whereas those with PSA below 1 ng/mL can safely be screened at extended intervals (e.g., 5+ years). This approach aims to reduce unnecessary procedures and overdiagnosis by lengthening the screening interval for low-risk men [6]. In parallel, the incorporation of multiparametric MRI into the diagnostic pathway—performing MRI in men with elevated PSA and only biopsying those with suspicious lesions—has been shown to improve the precision of screening by avoiding unnecessary biopsies and treatment for insignificant lesions [7].

At the policy level, there is growing support for population-based prostate cancer screening programs that implement these new strategies. In late 2022, the Council of the European Union updated its recommendations on cancer screening to include prostate cancer as a target, calling on member states to initiate organized pilot programs for PSA-based screening in conjunction with MRI as a second-line test [8]. This marked a significant shift in public health policy, as prior EU recommendations (from 2003) did not endorse prostate screening [9]. The new recommendation aligns with the EU’s Beating Cancer Plan and responds to persistent inequalities driven by opportunistic PSA testing (“wild screening”): in the absence of organized programs, more educated or proactive men tend to get PSA tests, while others (including high-risk groups) may miss out. Organized programs with standardized protocols and quality assurance are expected to improve early detection while controlling risks and ensuring equity.

Italy historically did not offer routine prostate cancer screening in its national health system, and guidelines were cautious about PSA testing, generally advising against screening asymptomatic men outside of clinical studies. However, in light of the new EU recommendations and emerging evidence, Italian regions have begun exploring organized screening models [10]. Lombardy—Italy’s largest region with ~10 million inhabitants—is at the forefront of this effort. In July 2024, the Lombardy Regional Government approved a structured pilot program for prostate cancer screening (DGR XII/2767/2024) [11]. This pilot was designed as a proof of concept to evaluate the feasibility, acceptance, and resource impact of a multilevel screening algorithm that leverages digital health infrastructure and risk stratification. The present article reports the initial results from the pilot’s activation phase (November 2024–June 2025) in Lombardy and discusses the strategic implications for scaling up the program.

The primary objectives of the pilot were to assess participation uptake, the risk profile of screened individuals, and the performance of the screening pathway (proportion needing further workup) in a real-world setting. Secondary objectives included identifying operational challenges (e.g., capacity strain on urology or imaging services) and gathering data to inform future expansions of the program. Ultimately, the goal is to use this evidence to guide an incremental rollout of organized prostate screening across Lombardy, aligned with international best practices and tailored to regional healthcare resources.

## 2. Materials and Methods

### 2.1. Program Design and Setting

This pilot screening program was conducted in the Lombardy Region of Italy and targeted men 50 years of age as the initial cohort. The pilot launch in late 2024 was intentionally limited to this youngest age within the eventual target range (50–69 years) to allow a controlled, stepwise implementation. The activation phase began on November 1, 2024, inviting men born in November 1974 (i.e., turning 50 that month), and subsequently added monthly birth cohorts: e.g., men born in December 1974 became eligible in December 2024, January 1975 births added in January 2025, and so forth. By June 2025, the pilot had been extended not only to all men born in 1974 (who were 50–51 years old in 2024–25) but also to those born in early 1975, effectively covering the entire 50-year-old population and some 51-year-olds. This corresponded to an eligible population of approximately 97,849 men.

Inclusion criteria: Men in the target age group (initially age 50) residing in Lombardy and enrolled in the regional health service were eligible to participate. There was no restriction based on sociodemographic factors; the program was open to all 50-year-old male residents, except for medical exclusions. The regional working group chose to start screening at younger ages because of the great potential gain of lifetime years for each case detected and the evidence of increased spontaneous screening at later ages.

Exclusion criteria: The screening protocol defined specific criteria to exclude individuals for whom PSA screening would be unnecessary or inappropriate. These were checked through both health records and self-reported data in a pre-screening questionnaire (see below). Temporary exclusions included (1) a PSA test performed within the last 2 years and (2) a prostate diagnostic procedure in the last 5 years, such as prostate biopsy, transrectal ultrasound, or abdominal CT/MRI imaging. Men meeting these criteria were deferred from the program because recent testing would obviate the need for immediate screening; they become eligible once the exclusion period (2 or 5 years) has elapsed. Permanent exclusions were (1) a history of prostate cancer (already diagnosed and under care) and (2) a known pathogenic mutation in a high-risk gene (e.g., BRCA1/2, CHEK2, ATM). Such individuals were not candidates for screening—those with prior prostate cancer would be in surveillance/treatment pathways, and known mutation carriers would be managed in high-risk protocols. Men who had any of these exclusions were identified via regional health databases (cancer registry, laboratory and hospital records) and by a questionnaire; if identified, they were not allowed to proceed to PSA testing.

### 2.2. Recruitment and Digital Portal Workflow

Passive invitation model: Unlike traditional screening programs, the pilot phase did not use mailed letters or active call-recall to invite individuals. Instead, an outreach campaign (through regional health websites, general practitioners, and media) informed the public that men turning 50 could access the screening on a voluntary basis via the online regional health record (the Fascicolo Sanitario Elettronico, FSE). Eligible individuals could log in with their personal credentials and find the prostate screening module available. This “pull” recruitment was chosen initially to gauge interest and manage demand within limited capacity. No personalized reminder letters were sent during the pilot.

Digital questionnaire and consent: Accessing the prostate screening module, the participants were prompted to complete a structured digital questionnaire (Appendix A) covering privacy consent, contact information, and screening eligibility questions. In Section 3 of the questionnaire, the users were asked to confirm the absence of the exclusion criteria: no previous prostate cancer, no PSA test in the last 2 years, no relevant diagnostic procedures in 5 years, and no known high-risk mutations. An answer of “Yes” to any exclusion question would terminate the process and classify the person as not eligible. If the answers indicated eligibility (or “No/Unknown” for the exclusion questions), the questionnaire then proceeded to Section 4 on family history (FH) of prostate cancer. The key question defined a positive family history as having a father, at least one brother, or at least one son diagnosed with prostate cancer. Men with a first-degree relative meeting that description who answered “Yes” were categorized as FH-positive; those answering “No” (or “I don’t know,” treated as no) were categorized as FH-negative. Family history was the only risk factor queried, as other factors (ethnicity, genetic status) were either homogeneous (the population is overwhelmingly Caucasian in Lombardy) or handled separately by exclusions.

Upon completion of the questionnaire, an immediate automated outcome was provided on the portal, with a PDF summary for the participant. Depending on the inputs, the system generated one of three outcomes:Not eligible: If any exclusion criteria were met, the participant was thanked and informed that he was not eligible for the screening program (with reasons, e.g., recent PSA test);Eligible, FH-negative: If eligible and with no positive family history, the participant was directed to proceed to blood testing for PSA and informed that further steps would depend on the PSA result (with no additional risk factors);Eligible, FH-positive: If eligible and family history was positive, the participant was likewise directed to get a PSA test, but also notified that having a familial risk factor already places him in a higher risk category.

For eligible participants, the portal provided an electronic order and authorization code for a free PSA test. The interface displayed a list of participating laboratories (all public and most accredited private labs in the region’s screening network) with a QR code and link, allowing the user to choose a convenient location. The PSA blood test did not require a physician’s prescription or payment (it was covered by the program). The participants could attend any listed lab with the code; the labs had been instructed on the protocol and used a dedicated test code to identify it as “PSA screening.” No pre-scheduled appointment was mandated; the men could walk in during lab hours for the blood draw, which contributed to the low-barrier, decentralized access model.

All data from the questionnaires and subsequent tests were captured in the region’s screening information system. The process was highly digitized, from consent and intake to result reporting, leveraging the FSE platform. The participants could choose their preferred communication method (email or phone text) for notifications regarding test results or next steps. They were informed that if further evaluation was needed, they would be contacted by their local health authority (ATS).

A schematic of the screening pathway is shown in Appendix B. The multilevel algorithm, largely derived from what was recommended at the European level by Van Poppel et al. [12], begins with an online questionnaire to assess inclusion criteria and familial risk, followed by a PSA test for eligible men. Subsequent management is based on the combination of PSA level and family history: FH-negative men with PSA < 1 ng/mL are classified as low risk and scheduled for routine rescreening after 5 years; FH-negative men with PSA between 1 and 3 ng/mL are considered intermediate risk and advised to repeat PSA testing in 2 years. Any man with PSA > 3 ng/mL or a positive family history is considered at elevated risk and referred for a urologic evaluation, including a digital rectal examination (DRE). Based on clinical findings, the urologist may recommend MRI (for intermediate or high clinical suspicion) or close follow-up. MRI findings then guide the indication for biopsy: typically, PI-RADS 4–5 or PI-RADS 3 lesions with high PSA density (≥0.10 ng/mL/cc) prompt biopsy. Men with negative evaluations (e.g., low PI-RADS and low PSA density) return to surveillance with periodic PSA testing, usually annually in higher-risk cases. This stepwise approach reflects current best practices to ensure that only men at elevated risk move on to intensive procedures.

### 2.3. Diagnostic Criteria and Follow-Up Procedures

#### 2.3.1. PSA Threshold and Risk Categories

The protocol stratified the participants based on PSA level in combination with family history:PSA < 1.0 ng/mL (with no FH): This was considered low risk. These men are very unlikely to have significant prostate cancer at age 50, based on evidence from risk modeling. They were scheduled for a long rescreening interval of 5 years (i.e., next screening at age 55), as per EAU recommendations and the pilot protocol. They were effectively exited from the immediate screening cycle and advised that no further action was needed until recall. For men with PSA < 1.0 ng/mL, a 5-year interval before re-testing was chosen based on longitudinal evidence showing very low prostate cancer risk in this group over a decade [13]. This extended interval reduces the burden of testing and minimizes overdiagnosis while maintaining safety.PSA 1.0–3.0 ng/mL (with no FH): This range was considered intermediate risk. These men were not referred immediately, but due to their higher PSA (above the age norm of ~1 ng/mL), they were at increased risk for developing detectable cancer in the near future. They were advised to have a shorter-interval recall with PSA testing in 2 years (at age 52). Thus, they remained in the program with closer surveillance.PSA > 3.0 ng/mL: This was the primary trigger for referral in PSA-based screening. A threshold of 3 ng/mL was chosen based on its use in major trials and guidelines as a level requiring further evaluation. All participants with PSA > 3.0 ng/mL were classified as screening-positive and were referred to a urology specialist for further evaluation. Importantly, this criterion applied irrespective of family history status. (In the planning, a prevalence of roughly 4–5% of 50-year-old men was expected to have PSA > 3 ng/mL).

The 3.0 ng/mL threshold was selected by the regional working group—comprising urologists, radiologists, epidemiologists, and patient representatives—based on its adoption in the ERSPC trial and endorsement in the European Association of Urology guidelines [12]. This cut-off will be subject to continuous reassessment as screening data accumulate.

The criterion of PI-RADS 3 in combination with PSA density ≥ 0.15 ng/mL/cm^3^ for recommending biopsy is consistent with European Association of Urology guidelines and is used in organized screening pilots such as PROBASE. Even the most recent literature recommendations support the use of PSA density to guide the diagnostic pathway [14]. This approach aims to balance sensitivity and specificity, reducing unnecessary biopsies while capturing clinically significant cancers.

Family history positive (any PSA level): The presence of a first-degree family history of prostate cancer was treated as an independent risk factor, warranting early evaluation. All FH-positive participants were referred to Urology, even if their PSA was modest. This recognizes that men with a significant inherited risk may merit closer examination (the protocol prespecified that these men receive a urologist’s input on further strategy). Notably, FH-positive men still underwent the PSA test; a few of them did indeed have high PSA as well, but even those with low PSA were sent for a urologic check. In practice, many FH-positive men with PSA below 3 were managed by urologists with advice to repeat PSA sooner (e.g., in 1 year) rather than immediate MRI, as per the protocol’s flexibility.

In our protocol, family history information is collected digitally and then validated during the initial urology consultation through direct patient interviews and clinical assessment, including a digital rectal examination. This step reduces recall bias and ensures more accurate risk classification.

#### 2.3.2. Urologic Assessment

Participants meeting referral criteria were contacted by their ATS to arrange a urologist consultation at a designated Screening Urology Clinic (typically in one of the public hospital networks). At this visit, the urologist reviewed the history, performed a DRE, and categorized the man’s risk status based on clinical findings and the PSA/family history context. The protocol defined three possible outcomes from the urologic exam:Low clinical risk: e.g., normal DRE, PSA only mildly elevated, and no concerning features. The urologist in this case could opt to defer invasive workup and advise repeating PSA at a shorter interval (such as 1 year). The participant would then be followed in the screening program with annual PSA checks until risk status changed.Intermediate risk: e.g., suspicious DRE or PSA persistently in the 3–10 ng/mL range without clear findings. The urologist would refer the participant for an MRI of the prostate to gather more information. No biopsy would be done at this stage; MRI results would determine the next steps.High risk: e.g., very high PSA (>10–15 ng/mL) or highly suspicious DRE. The urologist could directly indicate the need for MRI and a biopsy (anticipating a strong likelihood of significant cancer). Even in high-risk cases, MRI would generally be performed first to allow targeted biopsy of lesions.

Urologists also had discretion to exit someone from the screening program if they found a clear reason (e.g., a significant comorbidity or if the patient did not actually meet eligibility on closer review). They could also request a repeat PSA before deciding (for instance, if the initial PSA might have been elevated due to transient factors like prostatitis). These nuances were left to clinical judgment within the protocol guidelines.

#### 2.3.3. MRI and Biopsy

Multiparametric MRI of the prostate (3 Tesla, with contrast as needed) was performed for those referred by urologists. MRI scans were reported with PI-RADS v2 scoring. The regional protocol included detailed technical requirements for MRI acquisition and reporting (e.g., use of pelvic phased-array coils, radiologist training) to ensure high quality. MRI results were integrated with PSA density (PSA level divided by MRI-estimated prostate volume) to decide on biopsy:MRI PI-RADS 4 or 5 (highly suspicious for cancer): Biopsy indicated.MRI PI-RADS 3 (indeterminate) with PSA density ≥ 0.10 ng/mL/cc: Biopsy indicated.MRI PI-RADS 3 with PSA density < 0.10: Can defer biopsy, recommend close follow-up (e.g., recheck in 1 year).MRI PI-RADS 1–2 (no lesion or clearly benign) with PSA density < 0.20: No biopsy; patient returns to surveillance (screening recall at 1 year, given the discordance).MRI PI-RADS 1–2 but PSA density ≥ 0.20: Consider biopsy despite a “negative” MRI, as the high PSA density suggests possible diffuse or MRI-occult cancer.

It should also be noted that patients with a PI-RADS score other than 1 or 2 are, in any case, reassessed by a urologist before proceeding with the pathway.

Prostate biopsies, when performed, were done via transperineal approach with local anesthesia (per regional practice) and included both systematic cores and MRI-targeted cores (for those with lesions). Histopathology followed ISUP grading. Men found to have significant prostate cancer (Gleason Grade Group ≥ 2, or Grade 1, with volume warranting treatment) were referred to appropriate treatment pathways. Men with negative biopsies or low-risk findings were generally advised to continue surveillance (e.g., annual PSA, repeat MRI in a year for PI-RADS 3, etc., per urologist’s recommendation).

#### 2.3.4. Quality Assurance

The pilot was overseen by a regional multidisciplinary steering committee. Standard operating procedures were provided in technical appendices for each step (laboratory PSA methods and calibration, MRI acquisition minimum standards, biopsy and pathology reporting, etc.). Data were collected centrally for monitoring. Key performance indicators defined for the pilot included participation rate, proportion of screened individuals with results indicating the need for further diagnostic follow-up, compliance with referrals, and downstream diagnostic yield (cancers detected). An interim analysis was planned after the initial months to inform adjustments.

#### 2.3.5. Data Collection and Analysis

We analyzed data from the pilot program’s start (1 Nov 2024) through 30 June 2025 (the end of the pilot’s eighth month). All participant records in the screening database were extracted, including questionnaire responses, laboratory results, and follow-up statuses. Summary statistics of participation and outcomes were calculated at the regional level and broken down by the local health agency (ATS) to assess geographic variation.

Key metrics defined were:Uptake (participation rate): the proportion of the eligible population who initiated the screening (i.e., completed the questionnaire) during the pilot period. The denominator (~108,000) was the estimated number of men in the target birth cohorts residing in Lombardy.Eligibility rate: the proportion of participants who were confirmed eligible (i.e., did not have an exclusion) and proceeded to PSA testing.Risk factor positivity: the proportion of participants with a positive family history.PSA result distribution: percentages of screened men falling into the PSA-based risk categories (<1, 1–3, >3 ng/mL).Referral rate: overall percentage of participants who screened positive and were referred for urologic evaluation (this includes those with PSA > 3 ng/mL and/or FH positive).Findings at assessment: number of men undergoing urologic exam, and subsequent MRI and biopsy, as available in the data. (Given the short follow-up, cancer detection yield was expected to be very low; nonetheless, any confirmed diagnoses were recorded.)

Simple descriptive statistics (proportions, means) are reported. No formal hypothesis testing was applied, as this is primarily a process outcomes report. The analysis was conducted using Python/Pandas and spreadsheets provided by the regional screening coordination center. The data in this report were de-identified and aggregated, reflecting service delivery metrics.

Ethical approval was not required for this pilot program analysis because it was conducted as a public health initiative under regional policy mandate (DGR 2767/2024) and involved no experimental intervention beyond standard-of-care early detection. All participants provided informed consent electronically before entering the program. The study follows the principles of the Helsinki Declaration; confidentiality of personal data was maintained using secure regional health data infrastructure.

## 3. Results

### 3.1. Participation and Baseline Characteristics

#### 3.1.1. Population Reached

As of the end of June 2025, all men born in 1974 and in the first half of 1975 were in the age group identified for prostate cancer screening, totaling 123,736 individuals. Among them, 97,849 (79%) had no recorded PSA test in the previous two years in the regional databases. These 97,849 men were eligible to access the online questionnaire to participate in the screening program.

From November 2024 through June 2025, a total of 8558 men accessed the online portal and completed the screening questionnaire (Table 1). This corresponds to an approximate uptake of 8.7% of the 97,849 eligible men in the target age group during that period (men without record of a PSA test in the past two years in the regional database). Participation was entirely upon individual choice, as no individual invitations were sent. The monthly trend indicated a slow start in November (only a few hundred participants in the first 2 months) with modest increases after eligibility cohort enlargement in early 2025, but overall uptake remained low in this initial passive-invitation phase.

To further clarify the program’s activation process, Figure 1 combines two complementary views: panel A shows the chronological timeline of pilot rollout and monthly enrollment milestones, while panel B presents a Gantt chart detailing the monthly recruitment of birth cohorts and the planned cohort expansions in subsequent phases. Together, these visual summaries illustrate the phased invitation strategy, the progressive expansion of eligibility, and the integration of interim evaluations before each expansion step.

#### 3.1.2. Eligibility

Among the 8558 men who expressed interest by filling out the questionnaire, 6072 (70.9%) were confirmed eligible for screening (the remainder had exclusions). Specifically, 2486 men (29.1% of the participants) were temporarily deferred due to recent PSA or diagnostic exams (“previous screening” in the last 2–5 years), and a very small number (<1%) were excluded for a prior prostate cancer or known genetic mutation. The vast majority of exclusions were due to a recent PSA test: many men had undergone opportunistic PSA testing in the previous year or two and were thus asked to wait until 2 years had elapsed. These individuals were labeled “Richiamo ad anni successivi” (to be recalled in future years) in the system. Among the 6072 men who completed the questionnaire, 1412 (23.3%) underwent the PSA test, 3263 (53.7%) still had the opportunity to take it, and 1397 (23.0%) had missed the chance to do so free of charge within the screening program, as more than 90 days had passed since completing the questionnaire.

Table 1 summarizes participation by health jurisdiction. These figures reflect the pilot’s first eight months. All eight ATSs in Lombardy enrolled participants, roughly in proportion to their population size. ATS Milano (which includes the large metropolitan area) accounted for the largest share, with 3401 participants (39.7% of total), followed by ATS Insubria (which covers Varese/Como, 1.296–15.1%) and others. The smallest numbers came from the mountain area (ATS Montagna, 237 participants 2.7%, reflecting its smaller population). We did not observe extreme geographic disparities in uptake; however, overall numbers were too low for firm conclusions on regional variation. The participants’ mean age was 50 years (range 50–51 in this pilot, since it included 1974–1975 births). No other demographic data (education, ethnicity) were collected in this phase, but presumably, the cohort was predominantly Caucasian and Italian-born, reflecting Lombardy’s demographics. Additional analyses are planned to characterize the participants by urban versus rural residence, educational attainment, and deprivation index. This will enable targeted strategies to address gaps in participation across sociodemographic strata.

### 3.2. Screening Test Results and Risk Stratification

#### 3.2.1. PSA Testing Compliance

Out of the 6072 eligible men, by the data cutoff (end–June 2025), a subset had completed the test and received results, whereas others (especially those who joined in late May/June) were still pending their lab visit or result. A total of 1412 participants had obtained a PSA result by June 30. The remaining eligible men were in the process (either awaiting their lab appointment or result). We estimate that roughly one-third of the eligible cohort completed screening fully within the pilot window; many others who enrolled in May–June would complete their PSA in subsequent weeks. All PSA tests were processed in regional laboratories using standard immunoassays calibrated for total PSA (the protocol did not include free PSA measurements at this screening stage).

#### 3.2.2. PSA Level Distribution

The PSA results observed confirm a wide variation among individuals, with most values being low. Table 2 presents the breakdown of the first-level screening outcomes integrating PSA and family history:About 824 (58.4%) of screened men have PSA < 1.0 ng/mL, qualifying as low risk (family history negative in these cases). These men were directed to 5-year recall. In the pilot, this low-PSA group constituted roughly two-fifths of the participants, which aligns with international data that 40–50% of men in their late 40s to 50 have PSA below 1 [15,16];Approximately 364 (25.8%) men have PSA in the 1.0–3.0 ng/mL range (and FH-negative). This intermediate group—over half of participants—is slated for 2-year recall. The relatively large size of this group is expected, as PSA values between 1 and 3 are common at age 50. Many of these men’s PSAs cluster near the lower end of that range; only a minority were close to 3;Around 2.1% (n. 29) of men had PSA > 3.0 ng/mL (FH-negative) on their screening test, exceeding the referral threshold. This figure is consistent with the program’s planning assumption that ~15% of participants would be referred for further workup when combining PSA > 3 and FH risk (since ~13% have FH, ~2% would need to have PSA > 3 to total 15%). Indeed, as more results came in, the cumulative referral rate approached that target (see below);By definition, 100% of FH-positive men are considered elevated risk regardless of PSA. In the pilot, 195 (13.8%) participants were FH-positive. Their PSA distribution was variable: some had very low PSA (e.g., 0.4–0.6) but still went to urology due to family history, whereas a few had high PSA as well. The protocol treated FH-positive with PSA >3 equivalently, but practically these were double-flagged (they would have been referred for either criterion).

An important summary measure is that approximately 42% of the screened men had PSA > 1.0 ng/mL (the threshold that EAU guidelines consider “at risk”, requiring more frequent screening) or were FH-positive and conversely 58% had PSA ≤ 1.0 and were FH-negative. Thus, a majority of men screened will need a shorter rescreening interval (2 years), while a sizable minority (two out of five) can safely go to 5-year intervals. This risk stratification significantly reduces the intensity of screening for low-risk individuals.

### 3.3. Referral and Second-Level Findings

#### 3.3.1. Referral Rate

In total, approximately 15.9% of the screened men were referred to urologists based on the initial screen. This confirms that the workload for assessment (urology visits) is on the order of 1 referral per 6–7 screened men.

#### 3.3.2. Urologic Examination

Of the 224 referred men, 133 (59.4%) had completed their urologist visit by end of June and another 58 (25.8%) had an appointment scheduled or pending; 27 (12%) men had undergone the PSA visit, but the registration of the results was still pending. Six patients refused to undergo urological examination. The majority of referrals in May–June were still awaiting scheduling at the time of data cutoff, as the system was ramping up. Among the first 133 men seen by urologists, 16 had come via the PSA > 3 route and 117 via the FH-positive route, and only 3 also had PSA > 3. The outcomes reported by urologists were as follows:No concerning findings (low clinical risk): 122 men (91%)—they were advised to continue routine surveillance. In 110 (82.7%) of these cases, the urologist explicitly noted a plan for a repeat PSA in 1 year; in 12 cases, the urologist recommended a follow-up within one year.Indication for MRI (intermediate risk): 10 men (7%)—based on DRE or PSA dynamics, the urologist requested an MRI for further evaluation.Other outcomes: 1 man declined further workup despite recommendations (recorded as “refused further care”).

These numbers, though small, suggest that fewer that one in ten men who see the urologist undergo an MRI (10 of 133 who completed visits, excluding those with deferrals). This aligns with the expectation that many PSA elevations at this age will still need imaging to determine if biopsy is necessary.

#### 3.3.3. MRI Findings

By 30 June, only 7 participants had completed an MRI exam:-Five men were considered at low risk (PI-RADS 2 and low-risk PSA density) and were deferred to a PSA test in one year-Two men were referred for a biopsy

#### 3.3.4. Biopsies and Cancer Diagnoses

By the pilot mid-point, only 2 men had undergone a prostate biopsy as part of the program workup: one man was positive for cancer and one was negative. The observed biopsy rate in this early phase should be interpreted with caution, as it reflects only the first months of program operation and a limited number of eligible referrals. As more participants reach the diagnostic steps, these rates are expected to stabilize and may differ from the current preliminary figures. Thus, given the short follow-up and small numbers of patients who reached the biopsy stage, it is unsurprising that zero cancers were diagnosed in the first eight months. While the aim of this report is not to assess screening impact on mortality or overdiagnosis, this finding illustrates that the multilevel pathway can identify clinically relevant disease at an early stage. The European Randomized Study of Screening for Prostate Cancer (ERSPC) reports a number needed to screen of 781 for men aged 55 with 10-year follow-up [17], whereas in men aged 50, the cumulative incidence and mortality are lower, with an NNS estimated at approximately 2000. Prostate cancer prevalence at age 50 is relatively low, and the yield of screening in this initial round was expected to be modest. The pilot’s main focus was on process metrics; outcome metrics like cancer detection rate will require a longer observation period and more participants.

In aggregate, the pilot’s early results demonstrate the feasibility of the digital, multilevel screening approach, while revealing low uptake under a passive recruitment strategy. Among those who did participate, the majority were successfully triaged into appropriate risk categories: more than half (58%) were slated for routine 5-year follow-up, ~26% safely excused for 2 years, and 16% escalated to further evaluation. The digital platform functioned effectively to stratify and route patients—e.g., all referrals were automatically flagged and managed by ATS for appointments. Importantly, no major adverse events were reported; the PSA test itself led to few if any complications (one could expect <1% minor issues like vasovagal reactions with blood draw). No instances of overdiagnosis/overtreatment have occurred to date, as no low-risk cancers have been picked up and sent to surgery, for example—in fact, the challenge has been the opposite (ensuring enough high-risk cases move forward in a timely way).

Capacity-wise, the initial low volume meant that clinical services were not overwhelmed. Laboratory capacity for PSA tests was ample (only ~1412, tests done in 8 months, easily handled), and urology clinics saw referrals trickling in slowly (a few dozen at most in each ATS by April). MRI slots for the handful of patients indicated were readily available. However, these conditions reflect very low participation; they do not yet test the system’s limits. The pilot thus provided a stress test under low load but valuable insights for improvements before expanding scale.

#### 3.3.5. Projected Cancer Yield

Based on the observed participation rates, PSA distribution, and referral patterns in the first eight months, we modeled the expected prostate cancer yield under different participation scenarios and age ranges (Table 3). These estimates use regional incidence data and test performance assumptions from established screening trials.

For the current pilot cohort of 50-year-old men without a recent PSA test, the observed participation rate yields a very small number of cancers in the short term, as expected, given the low prevalence in this age group. Projections for broader age ranges (50–69 years) and higher participation rates show that even with moderate uptake (15–30%), the program could detect several hundred cases over a two-year cycle. Test sensitivity was set at 87%, and incidence rates were based on Cancer Registry data from ATS Milan (2019). These projections are intended as operational benchmarks to inform resource allocation, not as forecasts of mortality impact.

## 4. Discussion

In this pilot implementation of an organized prostate cancer screening program in Lombardy, we achieved proof of concept of a digitally driven, risk-stratified screening model, albeit with lower than hoped initial uptake. These results carry several implications for the future rollout of prostate screening, both within our region and in other jurisdictions considering similar programs.

### 4.1. Feasibility of the Multilevel Model

The screening algorithm—combining an online risk questionnaire, PSA testing, and selective referral to specialist evaluation—was executed successfully. The information technology infrastructure (regional e-portal and integrated data flows) performed well, demonstrating that a fully digital process is viable for guiding participants through screening. In essence, the program capitalized on existing e-health tools to automate what would otherwise require mailed invitations, paper forms, and manual scheduling. This suggests that large health systems with mature IT (like Lombardy’s FSE) can leverage these assets to implement screening in a paperless and patient-driven manner. We consider this a significant innovation, as it lowers administrative burden and provides participants with agency in managing their health. Moreover, stratifying risk at the entry point (via questionnaire) and again after PSA allowed efficient allocation of resources: only the subset at elevated risk proceeded to resource-intensive steps (urology visits, MRI). This pyramid approach aligns with recommendations from experts for risk-adapted screening pathways, and our operational data confirm its practicality in routine practice [19,20,21,22,23].

Lombardy’s implementation leveraged the European Digital Identity framework for secure access to the regional health portal. Technically, no additional software beyond that already in use for other organized screenings was required, and privacy management fully complied with the General Data Protection Regulation (GDPR). These infrastructural features may differ from those of other Italian or European regions, potentially influencing scalability.

### 4.2. Low Participation and the Need for Active Invitation

A salient finding is the ~8% participation rate observed under the passive enrollment approach. This uptake is very low compared to organized screening programs for other cancers in Lombardy (for example, colorectal screening often achieves ~50% uptake with mailed invitations). It is even far below the ~47% participation seen in the GÖTEBORG-2 randomized trial, which actively invited men 50–60 years old [4]. Several factors likely contributed: First, without personal invitations, many eligible men may simply be unaware of the program. Public communication efforts clearly did not reach or motivate the majority in this short window. Second, the pilot cohort was limited to 50-year-olds—an age group that had never been targeted for screening before; they and their primary care providers might not yet have a preventive mindset for prostate cancer (unlike, say, screening at screening at older ages). Third, some men might have been hesitant or preferred to consult their doctor first; since GPs were not formally integrated in inviting patients, this could slow uptake. Finally, opportunistic screening outside the program competes with the organized effort—indeed, ~30% of interested men were found ineligible due to recent PSA tests, indicating a substantial number are getting tested on their own or by physician advice. To improve participation, a shift to an active invitation system is planned. The regional authorities have decided that in subsequent phases, personalized invitation letters or electronic notices will be sent to each eligible man, as is done for other cancer screenings. These invitations will likely reference the availability of the online portal but serve as a direct prompt, which evidence suggests is crucial for uptake in organized programs [23]. We anticipate that with direct invitations and continued awareness campaigns (involving general practitioners and patient advocacy groups), the participation rate will rise substantially closer to the 30–50% range observed in trial settings and mature screening programs. Active invitation is recognized as a critical strategy for enhancing adherence within the Italian oncologic screening context. Evidence from both the initial phases of national screening programs and more recent years confirms that personalized invitations, including letters, substantially improve participation rates [24,25]. It is noteworthy that the Council of the EU’s recommendation explicitly calls for piloting new screening programs [8]—our experience underscores why initial operational kinks and public response must be evaluated and addressed on a small scale. The low response in our pilot served as a reality check, guiding us to adjust strategies (like adopting direct outreach earlier than initially planned). Importantly, the pilot’s slow uptake had a silver lining: it prevented any risk of overwhelming diagnostic services at the start. This gave us time to ensure that the care pathway functions properly for each referred individual.

### 4.3. Equity and Access

The implementation of organized, population-based screening programs has the potential to enhance equity in the provision of early cancer diagnosis, thereby reducing disparities in access. This is achieved through active invitation strategies, the free provision of diagnostic–therapeutic pathways, and targeted engagement approaches for disadvantaged populations [26,27]. Actions to address inequalities may include (1) conducting contextual analyses to identify causes and dynamics underlying disparities; (2) implementing general or tailored information campaigns; (3) improving service accessibility by adjusting hours or locations and deploying mobile units in rural areas; and (4) promoting mechanisms for social participation and empowerment.

In Lombardy, equity of access is addressed systematically through epidemiological surveys, which have shown that—despite the implementation of screening—sociodemographic determinants, chronic conditions, and barriers to accessing healthcare services remain significant predictors of non-adherence [28,29]. In addition, the region has developed tools such as the PRECEDE–PROCEED audit model, which has led to measurable improvements in the quality and equity of cancer screening programs [30,31]. For the activation phase of the prostate screening program, in-person access points with facilitators from the regional health service were established for individuals facing barriers to using digital tools. This multi-pronged approach aims to ensure that expansion of the program does not exacerbate existing inequalities and that vulnerable groups are actively supported in accessing screening services.

### 4.4. Risk Profile of the Screened Cohort

The distribution of PSA levels and risk factors among participants has implications for long-term outcomes and resource needs. We observed that more than half of men had PSA < 1 ng/mL at age 50, meaning they can likely be screened at much longer intervals without missing significant cancers. This supports the idea of extending screening intervals for low-risk individuals, which reduces burden and potential harms. In our program, we set a 5-year interval for PSA < 1; emerging evidence (e.g., the German PROBASE trial) suggests even a 5–8-year interval might be safe for PSA below 1.5 ng/mL in the mid-40s [32]. As we follow our cohort, we will evaluate if any aggressive cancers arise in men with PSA < 1 at baseline—the expectation is very few, validating the extended recall.

Conversely, the minority (40%) had PSA ≥ 1 and thus merit closer monitoring. Around 16% were flagged for immediate workups (PSA > 3 or FH). This fraction is crucial as it contains those most likely to harbor clinically significant cancer early. Our data suggest that by focusing on this ~16%, we can direct advanced diagnostics (MRI/biopsy) to the subset that truly needs it, sparing ~24% of men from unnecessary invasive procedures. For instance, none of the men with PSA < 1 in our pilot went to biopsy, and likely none had cancer that would have been detected via screening anyway at that time—a win for specificity. Meanwhile, among those referred to urology, a proportion will indeed have or develop significant cancer. Although no cancers were confirmed yet, one MRI (PI-RADS 4) strongly indicated a tumor (biopsy pending)—if that yields a diagnosis, it validates the pathway’s ability to find aggressive disease in its early stage.

### 4.5. Resource Implications and System Capacity

A strategic concern for any new screening program is whether the healthcare system can absorb additional diagnostic and treatment procedures. In Lombardy’s pilot, we closely monitored capacity utilization:Urology consultations: We projected that ~16% of the participants would need a urologist visit. At full target population (eventually all men 50–69 years old over the years), this could be a substantial volume. However, the plan is to phase into age cohorts gradually (as we started with 50-year-olds, then 51, etc.), so that each year only one new age group is added. This spreads the influx over two decades. Additionally, if ~40% of men never respond even after invitations (a realistic scenario), the effective referral load is lower. During the pilot, urology clinics easily handled the referrals (which were very few). The true test will come as participation increases. We are mapping out capacity: Lombardy has a robust urology network, but certain areas may need to allocate more clinic slots. One mitigation strategy is that not all referrals occur at once—they will trickle in throughout the year as people get screened in a staggered fashion (birthday-based). The regional coordination is also considering involving office-based urologists and training them in the protocol to expand capacity, as well as telemedicine consults for straightforward cases (e.g., discussing a plan for FH-positive men with low PSA could be done remotely).MRI imaging: An MRI was indicated in about half of those seen by urologists in our early data. If 16% see urology and ~7% of those get MRI, that implies ~1% of all screened men ultimately need an MRI. It should be noted, however, that the current sample consists of younger men; therefore, the reported percentages are expected to change over time as older age groups are progressively invited. Lombardy’s radiology services have finite MRI machines and slots, and many are already utilized for other indications. Therefore, MRI capacity is a potential bottleneck if the screening volume grows quickly. The pilot highlighted this issue, albeit on a small scale. One planned response is the adoption of abbreviated MRI protocols (e.g., biparametric MRI without contrast, ~15-min scans) for screening purposes, as suggested by the UK’s IP1-PROSTAGRAM study, which found that a short MRI could detect significant cancers with similar biopsy rates as PSA screening [33,34]. Implementing faster MRI exams dedicated to screening could increase throughput. The region’s technical guidelines (Appendix C) also set minimum requirements so that MRI can be done at more centers (including accredited private clinics) to boost capacity. Funding from the national oncology plan has been earmarked to invest in MRI services specifically for screening.Biopsies and downstream treatment: The fraction of men proceeding to biopsy is yet to be seen (likely a subset of those with positive MRI). Our protocol’s restrictive biopsy criteria (only PI-RADS ≥ 4 or PI-RADS 3 with high PSA density) aim to keep biopsy rates low, focusing on likely higher-grade cancers. In the pilot’s first months, biopsy utilization was extremely low (basically one case). Even with higher participation, we anticipate a biopsy rate well below that of historical PSA screening. For example, in ERSPC, many biopsies were done on PSA 3–4 with no MRI filter, whereas here MRI-negative men mostly avoid biopsy. Thus, while treatment facilities must be ready for an uptick in detected cancers, we expect the increase to be gradual and manageable. The absence of any overdiagnosed low-risk cancers so far is encouraging—none of the early detected lesions were low-grade (indeed we have none diagnosed yet; if the pending case is positive, it is likely intermediate or high grade given PI-RADS 4). Over time, we will track detection rates and ensure that any surge in cases can be handled by surgical and radiotherapy centers. The regional cancer network is involved in planning, to avoid treatment delays for newly diagnosed cases.

Regional Resolution XII/2767 of 15 July 2024 defined multiple implementation scenarios to ensure sustainability even with participation rates up to 40%. The program’s phased expansion—from a one-month birth cohort to one-year and then five-year cohorts—allows continuous assessment of system capacity. Current resources, supported by EUR 1.7 million annually from the National Oncology Plan in addition to routine regional funding (~EUR 20 billion healthcare budget), guarantee sustainability until at least 2027 at present uptake levels. If participation increases, additional resources can be mobilized immediately, and the program is expected to replace some services currently delivered opportunistically, improving efficiency.

At the current uptake, the cost per man screened is EUR 5.1, covering laboratory, urology, and imaging components. The first cancers detected correspond to an approximate cost of EUR 7143. These estimates will be refined as further outcome data become available, enabling a formal cost-effectiveness analysis. Early evidence also indicates that the program is significantly reducing the number of urology visits, thereby improving the overall efficiency of screening and prevention pathways. In Lombardy, the average screening costs are EUR 9835 per breast cancer case, EUR 527 per colorectal polyp or cancer, and EUR 1674 per CIN2+ cervical lesion. A cost of around EUR 7000 per prostate cancer case therefore falls well within the range already accepted by the regional public health service for other organized screening programs.

### 4.6. Comparison to Other Studies/Programs

Unlike other pilot projects, our initiative is not primarily designed to re-evaluate the effectiveness of PSA-based screening or its sustainability in terms of overdiagnosis. These issues have already been addressed through robust existing evidence and endorsed by both international (EU) and national (PON) recommendations. Instead, the distinctive aim of our program is to translate current scientific evidence into a tangible health opportunity for citizens, ensuring a sustainable access pathway, continuous analysis of results and the economic viability required for long-term implementation. Importantly, the design of the pilot avoids placing undue strain on the Lombardy healthcare system by preventing an increase in service demand beyond the capacity to provide a dedicated, high-quality response.

Our pilot shares similarities with and draws lessons from other international efforts:The Swedish and Dutch experiences (ERSPC centers, Göteborg) showed that organized PSA screening can reduce advanced disease and mortality [4,35]. However, they also highlighted overdiagnosis. By incorporating MRI and risk-based intervals, Lombardy’s model represents a modernized screening approach. Early data from the UK (BARCODE, PROSTAGRAM) and German PROBASE trials support this kind of approach [32,33,36]. Our real-world implementation provides practical evidence outside a trial setting.Lithuania’s national PSA screening program (cited as the only EU country with routine PSA screening so far) achieved significant coverage but without initial MRI triage [37]. Our results could inform them and others on how to integrate MRI and handle referrals efficiently.Notably, some regions/countries remain cautious (e.g., the UK’s NHS has not launched screening yet, though planning research pilots) [38]. The low uptake we saw might be used as an argument by skeptics that men are “not interested” in PSA screening. We interpret it differently: men will participate if properly invited and educated—hence our pivot to direct invitations. Also, men who did come forward in our pilot likely represent the more motivated segment; the challenge will be reaching those less proactively engaged (perhaps by involving GPs more intimately in recommending the program during health visits). This touches on the health equity aspect: we must ensure that all socio-economic groups are informed. Future expansions will include a communication strategy targeting lower-awareness populations, possibly through local public health units.

The regional working group opted to initiate screening at age 50 to maximize the potential lifetime years gained per case detected and in light of growing evidence of spontaneous PSA testing at older ages. Earlier initiation may capture aggressive cancers before they progress, while still allowing risk-based interval tailoring to avoid unnecessary interventions. Based on ERSPC and PROBASE data, simple modeling can be applied to estimate the number of cancers likely to be detected in this cohort at different time horizons. For example, assuming an age-specific incidence of 0.6% per year in men aged 50 and a biopsy positivity rate of 30% for MRI-targeted biopsies, the expected number of detected cancers in the first full year of screening at current uptake would be approximately 15–20. Such modeling will be updated annually as local data accrue, providing benchmarks for expected versus observed yields.

In this pilot, 2.1% of FH-negative men had PSA > 3.0 ng/mL—lower than the >10% observed in ERSPC 55-year-olds [19], yet higher than the <1% in the 45-year-old PROBASE cohort [32]. This difference may reflect the slightly younger age profile of our pilot cohort compared with ERSPC, as well as the self-selection of individuals with higher health literacy into this initial phase.

### 4.7. Strategic Adjustments and Future Perspective

Based on the pilot findings, several strategic decisions were made by the regional health authorities:Expansion of target ages.

Initially slated to add one birth year at a time, Lombardy accelerated the rollout. By June 2025 (sooner than first planned), the program was opened to all men aged 50–69 who had not had recent PSA tests. This was feasible because the pilot showed manageable volumes so far, and because waiting for more cohorts could delay benefits. Opening broadly may also boost participation by word of mouth and general practitioner engagement across all middle-aged patients. However, a cautious approach is maintained—if uptake surges, they may throttle invitations (e.g., invite by age groups in phases) to avoid overwhelming services.

2.Active invitation and reminders

Starting in the next few years, personalized invitations (via postal mail or electronic health record messages) will be sent to all eligible men who have not yet participated. If there is no response, periodic reminders may follow (similar to how colorectal screening sends a second letter). This is expected to dramatically improve adherence from the current ~3% towards the 30–40% range in the first round, with potential growth as the program normalizes. In organized cancer screening, it is common to see uptake increase over successive rounds as the population becomes familiar and trust is built.

3.General practitioner (GP) involvement

The pilot operated without formal GP recruitment, but feedback suggests that when GPs endorse the program, men are more likely to participate. Plans are underway to integrate GPs by providing them lists of their eligible patients and educational materials, so that they can actively recommend the screening during routine visits. GPs will also gain access to their patients’ screening status and results through the FSE professional portal, facilitating follow-up discussions.

4.Communication and education

The region recognized that knowledge gaps and fear of overdiagnosis might deter some men. A tailored communication campaign will accompany the next phase, emphasizing that the new screening is not the old one-size PSA testing: it is risk-based, with careful avoidance of unnecessary treatment (through MRI and active surveillance). Highlighting the free and user-friendly nature of the program and the potential to save lives by detecting aggressive cancer early will be important messages. Partnerships with patient advocacy groups (like Europa Uomo Italy) are being sought to spread awareness.

5.Monitoring and evaluation

The pilot results serve as a baseline. Going forward, a comprehensive evaluation framework is in place. Key performance indicators include participation rate overall and by demographics, positive predictive value of referrals (proportion of referred men found to have significant cancer), stage and grade of cancers detected (with an eye on detecting mostly clinically significant tumors), and interval cancer rate (cancers that present between screening rounds, which indicates false negatives). The program will be refined continuously—for example, if data show that PSA density threshold or MRI criteria could be adjusted to improve yield, or if an age to stop screening (our upper age is 69 by design) might be lowered or raised based on outcomes, etc.

For the early phases of implementation, interim evaluation will be based on process indicators, including the proportion of PSA-positive men undergoing urology consultation, the proportion of referred men completing MRI, and the percentage of MRIs rated PI-RADS ≥ 3. Diagnostic yield will be monitored as the proportion of biopsies performed and the proportion of cancers detected among those biopsied. These measures will inform operational adjustments while awaiting long-term outcome data.

The program will be linked with the Regional Cancer Registry to estimate sensitivity through interval cancer analysis, and staging of screen-detected cancers will be recorded to assess down-staging at the population level.

Linking process indicators to clinical outcomes will be a core element of program evaluation. The Regional Cancer Registry will enable analysis of tumor stage, grade, and risk category at diagnosis for screen-detected and interval cancers, allowing assessment of down-staging and shifts in disease severity over time.

### 4.8. Limitations

This report covers an early phase with limited follow-up; thus, we cannot yet draw conclusions on cancer detection rates or mortality impact. The uptake was low, so the participants may not be representative of the entire population (they might skew to more health-conscious individuals or those with family history, although our FH rate was only 10%). A selection bias is likely in this initial phase, as participants were self-selected under a passive invitation model. The main purpose at this stage, however, was to demonstrate the operational feasibility of the pathway, which is now being refined for broader, more representative participation. If so, as invitations go out, we might see a different risk profile in new participants. Also, many metrics (like the 40% > 1 ng/mL PSA figure) are derived from a subset or projected—those will firm up as numbers grow. Another limitation is that we did not survey participants about their experience; understanding why 92% did not participate is mostly conjecture at this point. As the program expands, conducting qualitative research on barriers (e.g., fear, lack of info, inconvenience) will be valuable to address hesitancy. Lastly, the pilot took place in a region with advanced healthcare infrastructure; results might differ in areas without such digital systems or with different population characteristics.

Despite these limitations, our findings offer valuable insights for any region or country implementing modern prostate screening. A key takeaway is the importance of combining PSA with additional risk markers such as family history and MRI—an approach that helped make our system more sustainable. In 2023, 16,993 PSA tests were performed on 50-year-old men in the Lombardy Region, leading to 4240 urology consultations (25%), 312 MRI scans (1.8%), and 57 biopsies (0.34%).

Preliminary data from the screening program suggest that the proportion of urology consultations drops to 18.5% when accounting for cases requiring only follow-up after the initial visit or those involving an MRI followed by a second consultation. MRI usage decreases to 1.2%, while the biopsy rate remains unchanged at 0.34%.

Overall, in the 50-year-old population, screening is associated with a 25.9% reduction in urology consultations and a 35% reduction in MRI usage, with no impact on the biopsy rate (Table 4).

It is important to note that standard care data do not capture services delivered in the private sector; therefore, urology consultations, MRI exams, and biopsies may be underestimated in the standard care arm. As a result, the actual number of procedures potentially avoided through organized screening is likely even greater than observed.

Although the biopsy rate remained unchanged at 0.34% in the pilot, it should be noted that overtreatment generally increases with age [39,40]. In the broader population aged 50–69 in Lombardy, the biopsy rate rises to 0.66% relative to the number of PSA tests performed. These trends highlight the importance of tailoring screening protocols by age and reinforce the need for careful follow-up in older age groups to avoid unnecessary interventions.

Additionally, the need for adequate funding and planning is evident: Lombardy allocated a dedicated budget (~EUR 0.74 million from a national oncology fund) for this pilot and initial rollout, used for IT development, coordination, and covering the costs of PSA tests and further exams (which are offered free to participants). Sustained funding is crucial, as the program scales up; otherwise, cost pressures (particularly from MRI and biopsy volumes) could jeopardize it. Cost-effectiveness analyses will eventually be needed, incorporating our real data on detection and treatment.

On the policy front, our pilot answers the EU’s call for member states to experiment and build evidence on organized prostate screening [8]. We intend to share our results with national and European bodies. In Italy, other regions are closely watching Lombardy’s experience; some (e.g., Piedmont, Veneto) are contemplating their own pilot programs. The Italian Association of Urology and Oncology (AIURO and AIOM) is updating national guidelines, likely to cautiously endorse organized screening with safeguards (the 2024 AIOM guideline was under revision at the time of this writing). Our data will contribute to that discourse, showing that such a program can be implemented safely and that men are willing to participate when given a structured option.

Next steps and long-term outlook: By the end of 2025, we expect to have invited the entire 50–55 age band in Lombardy, with progressively improving uptake. Within a few years, the program should encompass all men 50–69 years old. As the rounds continue, we will accumulate outcome data—detection rates, tumor characteristics, and interval cancers—that will allow us to model the mortality impact. If the program is successful, we anticipate seeing a stage shift (more cancers detected at localized stages, with fewer metastatic cases arising) in our population over 5–10 years. This will be the ultimate measure of benefit, in line with trial evidence. It will also be important to monitor for overdiagnosis: we will track how many low-risk cancers (Gleason 3 + 3) are being picked up. Our protocol’s emphasis on not biopsying low-suspicion cases should minimize overdetection, but vigilance is required to ensure that we do not treat harmless tumors unnecessarily. The integrated use of active surveillance for low-grade cancer is part of the program’s continuum—any low-risk cancers that are found will be managed conservatively according to guidelines, further reducing harm.

These early findings suggest that the activation of Lombardy’s prostate cancer screening model represents a promising path forward for population-based prostate cancer control. The pilot affirms that a personalized, multilevel screening approach is logistically achievable and can efficiently stratify men by risk. While initial participation was modest, the lessons learned are already shaping improvements (transitioning from opportunistic to organized invitations). With these adjustments, the program is poised to expand and potentially serve as a prototype for national policy, aligning with European recommendations. Ongoing evaluation will determine if the anticipated benefits—reduced prostate cancer mortality and advanced disease, attained with acceptable harms and costs—materialize in the real-world setting. If successful, Lombardy’s experience could inform wider adoption of responsible prostate screening, turning a new page on a debate that has lasted for decades, and ultimately improving men’s health outcomes through earlier diagnosis of clinically significant prostate cancer.

## 5. Conclusions

The pilot phase of Lombardy’s organized prostate cancer screening program confirms the real-world feasibility of a risk-adapted, multilevel model built on digital infrastructure, PSA testing, and clinical stratification. Despite an initially modest uptake (~8%) under a passive invitation strategy, the program successfully identified and referred higher-risk individuals (16%) while minimizing unnecessary interventions in low-risk participants—an encouraging sign of appropriate targeting.

Key strengths included efficient digital enrollment, seamless integration with laboratory services, and adherence to risk-based clinical pathways involving MRI and biopsy only when indicated. Notably, no evidence of overdiagnosis or overtreatment emerged during this early phase. These process outcomes support the model’s sustainability and scalability, especially as strategic enhancements—such as active invitations, general practitioner engagement, and public communication—are implemented to improve participation rates.

As the program expands to encompass older age groups and a broader population (50–69 years), ongoing evaluation will be essential to assess its long-term clinical impact, including cancer detection rates, tumor characteristics, stage shifts, and potential mortality benefits. Monitoring of biopsy rates—especially in older men—and adherence to active surveillance protocols will be crucial to prevent overtreatment.

Importantly, this initiative aligns with the 2022 EU Council recommendations and may serve as a replicable model for other Italian regions and European countries aiming to implement evidence-based, equity-focused prostate cancer screening. Lombardy’s experience demonstrates that personalized, population-level screening is both technically viable and clinically sound when guided by current best practices. With sustained investment, transparent evaluation, and responsive adaptation, this program has the potential to reduce prostate cancer mortality while safeguarding men from unnecessary harm—marking a critical step forward in modern cancer prevention.

## Figures and Tables

**Figure 1 healthcare-13-02041-f001:**
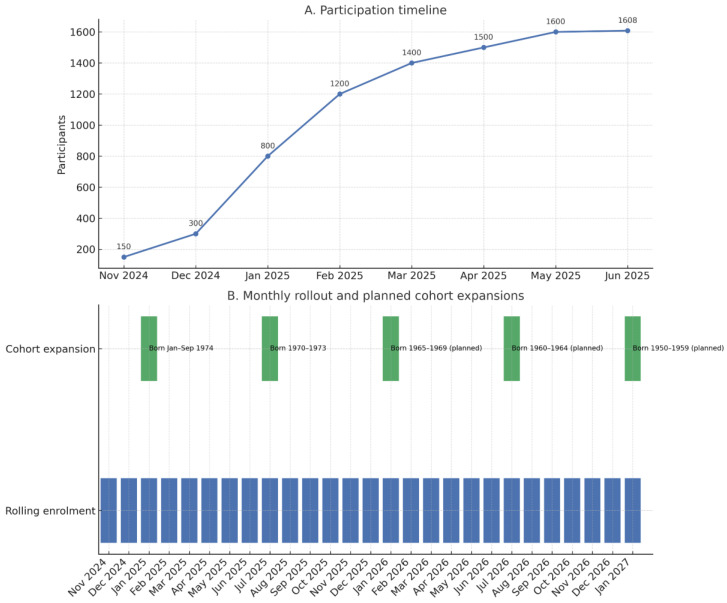
Pilot program rollout: participation timeline (**A**) and cohort invitation schedule with planned expansions (**B**).

**Table 1 healthcare-13-02041-t001:** Pilot participation by health agency (Nov 2024–Jun 2025).

Health Agency (ATS)	Participants (n)	Eligible for PSA (n)	Temporarily Ineligible ^1^ (n)
ATS Milano	3401	2366	1035
ATS Insubria	1296	919	377
ATS Montagna	237	169	68
ATS Brianza	1184	782	402
ATS Bergamo	624	508	116
ATS Brescia	910	680	230
ATS Val Padana	595	431	164
ATS Pavia	311	217	94
Total	8558	6072	2486

^1^ “Temporarily ineligible” refers to men who met exclusion criteria such as recent PSA test, and their eligibility date is rescheduled accordingly.

**Table 2 healthcare-13-02041-t002:** Stratification of initial screening results (pilot cohort).

Category (Screening Outcome)	N (% of Participants)	Recommended Action
FH-negative, PSA < 1.0 ng/mL (low risk)	824 (58.4)	Routine recall in 5 years
FH-negative, PSA 1.0–3.0 ng/mL (intermediate)	~364 (25.8)	Shorter recall in 2 years
FH-negative, PSA > 3.0 ng/mL (elevated PSA)	~29 (2.1)	Refer to Urology now (DRE)
FH-positive (any PSA) (elevated familial risk)	~195 (13.8)	Refer to Urology now (DRE)
Total participants screened	1412 (100%) ^1^	–

^1^ Percentages may not sum exactly to 100 due to rounding and overlapping criteria (a few men had both FH+ status and PSA > 3, counted in the FH-positive row only).

**Table 3 healthcare-13-02041-t003:** Projections of expected cancer yield from PSA screening in Lombardy.

Cohort (Age)	Eligible Population (No PSA Test in Past 2 Years)	Participation Rate (Scenario)	Number Screened	Expected Incidence (/100,000/Year) *	Interval (Years)	Test Sensitivity	Expected Cases	Expected Yield (/1000)
50 years–current situation	1412		1.4	21.1	2.000	0.870	0.5	0.4
50–69 years	861,163	15.0%	129,174	97.600	2.000	0.870	219.4	1.7
50–69 years	861,163	30.0%	258,349	97.600	2.000	0.870	438.7	1.7
50–69 years	861,163	40.0%	344,465	97.600	2.000	0.870	585.0	1.7

* Estimates based on ATS Milan Cancer Registry incidence data (2019) and ERSPC sensitivity assumptions [18].

**Table 4 healthcare-13-02041-t004:** Impact of prostate cancer screening vs. standard care in 50-year-old men (Lombardy Region, 2023).

Outcome Per PSA Test	Standard Care (n/N, %)	Screening Pilot (n/N, %) *	Absolute Difference (%)	95% CI
Urologist visits	4240/16,933 (25.04%)	261/1412 (18.48%)	6.56%	4.43–8.68%
MRI usage	312/16,933 (1.84%)	16.8/1412 (1.19%)	0.65%	0.05–1.25%
Biopsy rate	57/16,933 (0.34%)	4.8/1412 (0.34%)	−0.00%	−0.32–0.31%

* In order to compare the data from the actual annual production in Lombardy with those from the screening pilot and to avoid overestimating the screening’s capacity to reduce activities, the figures for urology visits, MRIs, and biopsies were adjusted in proportion to the number of PSA tests for which the screening process had not yet been completed (this explains why some figures are not absolute). In addition, for urology visits, those required in cases where patients need follow-up without an MRI scan after the initial consultation, as well as cases where an MRI scan is performed followed by a second urology visit, were also included.

## Data Availability

The data presented in this study are available on reasonable request from the corresponding author and the Regional Screening Coordination Center of Lombardia. Individual-level data are not publicly available due to privacy regulations. Aggregate results are reported in this article, and additional summary data can be obtained upon request.

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
