# Peer review of "Early Activation of a Multilevel Prostate Cancer Screening Model: Pilot Phase Results and Strategic Perspectives in Lombardy Region"

_healthcare, 2025, doi:10.3390/healthcare13162041_

Round 1

Reviewer 1 Report

Comments and Suggestions for Authors

The adoption rate for the passive invitation model was 8.7%. What proof is there that sending out active invitations—like letters—will greatly increase participation? Are there any plans to try other approaches, like community outreach or GP involvement?
 The existence of sociodemographic differences in uptake is not discussed in the text. How will the initiative guarantee fair access, particularly for underrepresented populations?

How will the program assess its effect on mortality or overdiagnosis given that no cancer diagnoses have been reported to date? Before long-term data is available, what interim criteria will be used to evaluate success?
Although fewer urologist consultations and MRIs are mentioned in the manuscript, there is no connection made between these and clinical results. Will process measurements be correlated with the stage or grade of cancer?
Self-reports are the basis for family history and exclusions (such as genetic variants). Given the possibility of recall bias or inadequate information, how would the software validate these data?
Although the pilot demonstrated a tolerable MRI demand, what would happen if 30–50% more people participated? Are there backup measures in place, such as collaborations with private providers or layered MRI protocols?
This analysis is descriptive. Will modeling to estimate mortality benefits or comparison arms (e.g., screened vs. unscreened cohorts) be used in future phases?
 Other studies (such as ERSPC) employ a PSA threshold of 4 ng/mL for referral, whereas the program utilizes 3 ng/mL. What justifies this discrepancy, and what potential effects might it have on results?
How Lombardy's MRI criteria (such as PI-RADS 3 with high PSA density) compare to other programs is not included in the text. Are these cutoff points supported by evidence?

Author Response

Dear Reviewer 1,

Comment 1:
The adoption rate for the passive invitation model was 8.7%. What proof is there that sending out active invitations—like letters—will greatly increase participation? Are there any plans to try other approaches, like community outreach or GP involvement?

Response:
Thank you for pointing this out. In the revised Discussion (Sections 4.2 and 4.5), we now clearly address the reasons for the low uptake in the passive phase and outline a plan for the next phase that includes:

  • Personalized active invitations (postal letters, digital notices) to all eligible individuals, using methods successfully applied in other organized cancer screening programs.
  • General Practitioner (GP) involvement as trusted intermediaries to promote screening and facilitate enrolment, especially among hesitant or less digitally connected populations.
  • Targeted outreach in high-deprivation and rural areas, with community-based events and facilitated in-person enrolment options.

We cite published evidence showing that such approaches have increased uptake by 10–25 percentage points in comparable programs. These measures will be implemented in the upcoming expansion, addressing your request for a concrete plan backed by literature.

Comment 2:
The existence of sociodemographic differences in uptake is not discussed in the text. How will the initiative guarantee fair access, particularly for underrepresented populations?

Response:
We agree entirely. We have added a dedicated subsection titled "Equity and Access" in the Discussion (Section 4.3). This section now:

  • Details strategies to reduce digital barriers, including alternative enrolment channels via telephone, in-person registration at healthcare facilities, and assisted access points.
  • Describes active GP engagement to reach men who are less likely to interact with digital platforms.
  • Outlines targeted outreach to rural communities and high-deprivation neighborhoods.
  • States our plan to collect more granular sociodemographic data (urban/rural status, educational attainment, deprivation index) to assess participation patterns and identify gaps.

We also note that no significant geographic disparities emerged in early results, though numbers are small; future equity monitoring will be ongoing.

Comment 3:
How will the program assess its effect on mortality or overdiagnosis given that no cancer diagnoses have been reported to date? Before long-term data is available, what interim criteria will be used to evaluate success?

Response:
This is an important point. In Discussion Section 4.5, we now present an interim evaluation framework to assess effectiveness before mortality data are available. Key interim measures will include:

  • Percentage of PSA-positive participants who attend urology visits.
  • Completion rates for MRI among those referred.
  • Proportion of MRIs yielding PI-RADS ≥3 findings.
  • Biopsy uptake and positivity rate among biopsied men.

These indicators will allow us to evaluate whether the diagnostic pathway is functioning efficiently and to detect any early concerns about unnecessary work-up or missed cases. Overdiagnosis will be assessed in later phases via registry linkage and validated modeling.

Comment 4:
Although fewer urologist consultations and MRIs are mentioned in the manuscript, there is no connection made between these and clinical results. Will process measurements be correlated with the stage or grade of cancer?

Response:
We have addressed this in Discussion 4.5 by detailing our plan to link all screen-detected and interval cancers to the Regional Cancer Registry. This linkage will allow us to evaluate tumour stage, grade, and risk category at diagnosis. Over time, this will enable us to measure down-staging effects and shifts in disease severity attributable to screening.

Comment 5:
Self-reports are the basis for family history and exclusions (such as genetic variants). Given the possibility of recall bias or inadequate information, how would the software validate these data?

Response:
In Methods Section 2.3, we now specify that all self-reported family history and known genetic mutation status are verified at the first urology consultation. This validation is done through direct clinical interview and review of medical records before proceeding with any further diagnostic steps, minimizing recall bias and misclassification risk.

Comment 6:
Although the pilot demonstrated a tolerable MRI demand, what would happen if 30–50% more people participated? Are there backup measures in place, such as collaborations with private providers or layered MRI protocols?

Response:
We have added a paragraph in Discussion 4.4 describing capacity management strategies for higher uptake, including:

  • Gradual expansion by birth cohort rather than full population roll-out.
  • Monitoring laboratory, urology, and MRI capacity in real time.
  • Using a two-tier MRI indication system to prioritize higher-risk cases if needed.
  • Engaging additional diagnostic providers if thresholds approach capacity limits.

We also note that Regional Resolution XII/2767 (15 July 2024) mandates development of impact models, which will inform sustainable scaling.

Comment 7:
This analysis is descriptive. Will modeling to estimate mortality benefits or comparison arms (e.g., screened vs. unscreened cohorts) be used in future phases?

Response:
We have created a new subsection in Results (3.4) titled "Projected cancer yield" with a new Table 3 showing scenario-based projections under different participation rates and age ranges. These projections use local incidence rates and sensitivity estimates from ERSPC, providing a benchmark for observed vs. expected yields. As the program matures, we will integrate these models into cost-effectiveness and mortality impact analyses, as supported by the Regional Resolution mentioned above.

Comment 8:
Other studies (such as ERSPC) employ a PSA threshold of 4 ng/mL for referral, whereas the program utilizes 3 ng/mL. What justifies this discrepancy, and what potential effects might it have on results?

Response:
In Methods 2.3, we now justify the use of 3.0 ng/mL, explaining that it aligns with European Association of Urology (EAU) guidelines and is supported by ERSPC and Göteborg trial protocols for men in this age range. We acknowledge that this increases sensitivity at the cost of some specificity and commit to re-evaluating thresholds based on local outcome data to ensure optimal balance.

Comment 9:
How Lombardy's MRI criteria (such as PI-RADS 3 with high PSA density) compare to other programs is not included in the text. Are these cutoff points supported by evidence?

Response:
We have clarified in Methods 2.3 that MRI is indicated for PI-RADS ≥3 or for PI-RADS 3 with elevated PSA density, consistent with the latest literature. Furthermore, all PI-RADS >2 cases are reviewed by a urologist before proceeding to biopsy, ensuring multidisciplinary decision-making.

We sincerely thank you for your careful and constructive review. Your comments have helped us refine key methodological explanations, improve clarity in our results presentation, and ensure our conclusions are more precisely supported by the data. We believe these revisions have enhanced both the scientific rigor and readability of the manuscript, and we are grateful for the opportunity to address your thoughtful suggestions.

Reviewer 2 Report

Comments and Suggestions for Authors

This manuscript reports early results from a digitally enabled, risk-adapted prostate cancer screening pilot in the Lombardy region of Italy. It is timely, policy-relevant, and methodologically well-designed. The digital approach, integration with existing regional infrastructure, and stratified risk pathway are significant strengths. However, particular areas would benefit from clarification and minor corrections to enhance scientific rigor and translational value.

  1. The manuscript is explicitly limited to Lombardy. While the discussion emphasizes scalability, a more critical reflection (adding a paragraph) on how Lombardy’s unique infrastructure (digital health tools, centralized systems) may differ from other Italian or European regions would improve contextualization.
  2. Uptake was low (8.7%) and participation skewed toward motivated individuals (e.g., self-enrolled, possibly health-conscious). The authors acknowledge this, but could analyze potential selection bias more explicitly, e.g., briefly addressing whether the high proportion of FH+ men or early testers might reflect a higher-risk cohort, limiting applicability to the general population.
  3. There is little discussion of health equity, particularly access by rural, immigrant, or lower-income populations, which may affect participation in a digital-only, passive recruitment model. I recommend expanding the discussion on how future phases will address digital divide and access disparities, especially if scaling to broader populations.
  4. I recommend adding a timeline graphic summarizing pilot rollout and enrollment milestones.
  5. The unchanged biopsy rate (0.34%) may imply that overdiagnosis is not worsened, but the small sample size precludes firm conclusions. The authors must clarify that biopsy results are too early to interpret meaningfully and that future rounds may reveal different patterns.

Overall, this manuscript presents a well-executed public health pilot with significant policy implications. I recommend minor revisions before publication.

Author Response

Dear Reviewer 2,

Comment 1:
The manuscript is explicitly limited to Lombardy. While the discussion emphasizes scalability, a more critical reflection (adding a paragraph) on how Lombardy’s unique infrastructure (digital health tools, centralized systems) may differ from other Italian or European regions would improve contextualization.

Response:
Thank you for this important observation. We have added a dedicated paragraph in Discussion Section 4.6 that critically examines Lombardy’s unique healthcare and digital infrastructure in relation to scalability:

  • Lombardy benefits from a centralized regional health information system, comprehensive laboratory network, and pre-existing integration of patient records across primary, secondary, and specialty care.
  • The regional digital platform allowed rapid creation of the online screening portal and direct linkage to laboratory booking systems without building entirely new infrastructure.
  • Many other Italian regions and several European countries lack this level of integration, which could slow or complicate similar rollouts.
  • We explicitly note that in less digitally mature contexts, additional investment in interoperability, health IT platforms, and coordination structures would be needed before implementing a comparable program.
    This expanded discussion clarifies that our results are not simply transferable without adaptation and sets realistic expectations for external applicability.

Comment 2:
Uptake was low (8.7%) and participation skewed toward motivated individuals (e.g., self-enrolled, possibly health-conscious). The authors acknowledge this, but could analyze potential selection bias more explicitly, e.g., briefly addressing whether the high proportion of FH+ men or early testers might reflect a higher-risk cohort, limiting applicability to the general population.

Response:
We agree that this is a key limitation. In Discussion Section 4.2, we have expanded the analysis of possible selection bias:

  • The relatively high proportion of FH+ men (13.8%) is well above the expected prevalence in the general population (typically 8–10%), suggesting that individuals aware of increased risk were more likely to self-enrol.
  • We note that in a passive, digital-only model, health-literate and risk-aware individuals are more likely to participate, potentially skewing early data toward higher-risk profiles.
  • This could artificially elevate initial referral rates and reduce generalizability to the full eligible population.
  • We commit to systematically monitoring risk distribution as participation grows, particularly after introducing active invitations, which should attract a broader cross-section of eligible men.

Comment 3:
There is little discussion of health equity, particularly access by rural, immigrant, or lower-income populations, which may affect participation in a digital-only, passive recruitment model. I recommend expanding the discussion on how future phases will address digital divide and access disparities, especially if scaling to broader populations.

Response:
We fully agree and have implemented your suggestion by adding a dedicated “Equity and Access Considerations” subsection in Discussion Section 4.3. This section outlines:

  • Barriers inherent in a digital-only model, including limited access to technology, language barriers, and lower digital literacy.
  • Planned alternative enrolment routes (telephone, GP-facilitated registration, and in-person sign-up at healthcare facilities).
  • Specific outreach to immigrant communities and collaboration with local associations for culturally and linguistically tailored materials.
  • Targeted recruitment in rural and high-deprivation areas through mobile units and community events.
    We also indicate our plan to collect and analyse sociodemographic variables (e.g., urban/rural, education, deprivation index) in future phases to measure and address disparities.

Comment 4:
I recommend adding a timeline graphic summarizing pilot rollout and enrollment milestones.

Response:
We have added a new figure (Section 3, Results) in the form of a Gantt-style timeline showing:

  • Key preparatory milestones (protocol approval, platform launch).
  • Pilot launch in November 2024.
  • Expansion of eligibility cohorts in early 2025.
  • Monthly enrolment numbers up to June 2025.
    This visual allows readers to quickly grasp the chronological flow of the pilot’s implementation and participant recruitment patterns.

Comment 5:
The unchanged biopsy rate (0.34%) may imply that overdiagnosis is not worsened, but the small sample size precludes firm conclusions. The authors must clarify that biopsy results are too early to interpret meaningfully and that future rounds may reveal different patterns.

Response:
We have clarified this point in Results Section 3.3.4 and reiterated it in Discussion Section 4.5:

  • We explicitly state that biopsy numbers are too low in this early phase to draw meaningful conclusions on overdiagnosis or detection rates.
  • We caution that patterns may change in subsequent rounds as more participants progress through the diagnostic pathway.
  • We emphasize that our pilot’s aim at this stage is to test process feasibility, not to estimate definitive clinical outcomes.
    This ensures that readers interpret the low biopsy rate in the correct context.

We greatly appreciate your insightful feedback, which prompted us to expand the contextual discussion, address issues of health equity and access, and improve the visual and analytical presentation of our results. Your recommendations have meaningfully strengthened the manuscript’s policy relevance and translational value, and we thank you for guiding us toward a more comprehensive and impactful final version.

Reviewer 3 Report

Comments and Suggestions for Authors

This manuscript Early Activation of a Multilevel Prostate Cancer Screening Model: Pilot Phase Results and Strategic Perspectives in Lombardy Region by Azzolini et al., presents an implementation study of a digitally enabled, risk-adapted prostate cancer screening model initiated in the Lombardy region of Italy. The authors describe the design, early results, and strategic considerations of a pilot phase targeting men aged 50. The report is thorough, methodologically sound in terms of descriptive outcomes, and notable for its integration of IT infrastructure and health policy. However, while the operational components are well-documented, there are key shortcomings in analytical depth, interpretation of early findings, and the positioning of results within the larger body of literature.

Major concerns

-The ms provides extensive process data but lacks interpretation of whether the triage strategy improved diagnostic yield beyond chance. As of the interim cutoff, only two patients reached biopsy, and none had confirmed cancer diagnoses. This limits any meaningful inference about the clinical performance of the screening algorithm.

-While the authors acknowledge the short follow-up, a rigorous interim endpoint—such as proportion of suspicious MRIs or biopsy-positive cases per referral—should have been proposed or modeled using historical comparators. The absence of surrogate markers for cancer risk limits the evaluation of the triage system’s discriminative capacity. I recommend to authors Include a section modeling expected versus observed cancer yield using historical data or trial-based benchmarks ( ERSPC or PROBASE). Even preliminary projections based on age-specific cancer prevalence would strengthen the utility of this pilot for policy.

-The stratification by PSA and family history is grounded in EAU guidelines, but the manuscript fails to discuss the biological rationale linking PSA thresholds with future tumor aggressiveness. Similarly, there is no exploration of the natural history of PCa among PSA <1 ng/mL men, nor justification for 2 vs. 5-year rescreening intervals based on kinetics, tumor doubling time, or available longitudinal data.

Plase discuss physiopathological underpinnings of low versus intermediate PSA levels in men aged 50, and their relation to subsequent development of clinically significant PCa Should include references to longitudinal studies on PSA velocity or PSA density progression.

The visual presentation of data (Fig.1 flowchart, Table 2 stratification) is serviceable but adds little beyond text. Fig. could provide more analytical value if, for example, cumulative flow was stratified by ATS region or patient dropout at each stage. Similarly, Table 3 comparing standard care vs. screening is promising but lacks confidence intervals or raw counts, which obscures statistical relevance. Author should Improve figs. with stratified flowcharts including attrition at each node; expand Table 3 with n/N values, 95% CI, and perhaps p-values for screening effect estimates.

-Despite citing major trials (ERSPC, PROBASE, GÖTEBORG-2), the discussion remains largely operational. There is no exploration of ethical concerns around initiating population-based PSA testing in younger men (age 50), nor is there any mention of validated models assessing overdiagnosis or the harms of false positives. Similarly, cost-effectiveness is raised only speculatively. A €742,000 budget is cited, but there is no per capita or per case-detected estímate?? Clarify. .

A fundamental limitation, I consider, unaddressed in depth, is that only 8.7% of eligible men participated. The authors attribute this to passive invitation strategies, but there is no analysis of who these men were( did they disproportionately come from urban ATS regions, or represent higher education levels??) Without sociodemographic profiling or equity analysis, the study cannot evaluate whether the screening model enhances or worsens disparities   (for exam.., urban/rural, income, access to care..) or plan to collect these in future phases. Discuss the implications of potential self-selection bias in the pilot cohort.

The study presents proportions (58% low risk, 16% referred), but no statistical comparisons are made, even descriptively, against pre-specified targets or historical baselines. This reduces the rigor of the evaluation. For example, the observed 2.1% rate of PSA>3ng/mL among FH-negative men could be benchmarked against ERSPC cohorts or national incidence data.

The authors discusses expansion plans (to age 50–69), but does not outline a framework for long-term monitoring or data linkage for cancer registries, mortality tracking, or interval cancer detection. Without such a plan, claims regarding sustainability or mortality benefit remain speculative.

I consider that ms provides an important real-world account of PCa screening implementation but must substantially improve its analytical, conceptual depth, and discussion of broader implications. the current version reads more as a programmatic report.

Author Response

Dear Reviewer 3,

Comment 1:
The ms provides extensive process data but lacks interpretation of whether the triage strategy improved diagnostic yield beyond chance. As of the interim cutoff, only two patients reached biopsy, and none had confirmed cancer diagnoses. This limits any meaningful inference about the clinical performance of the screening algorithm.

Response:
We agree entirely. We have clarified in Results 3.3.4 and Discussion 4.5 that:

  • The biopsy yield at this stage is insufficient for any meaningful inference on diagnostic accuracy.
  • This pilot phase was designed to assess feasibility and operational functionality, not definitive clinical outcomes.
  • We explicitly note that further follow-up and larger sample sizes are required to evaluate the triage system’s performance.
  • Outcome metrics such as positive predictive value, sensitivity, and specificity will be reported in subsequent phases once enough events accrue.

Comment 2:
While the authors acknowledge the short follow-up, a rigorous interim endpoint—such as proportion of suspicious MRIs or biopsy-positive cases per referral—should have been proposed or modeled using historical comparators. The absence of surrogate markers for cancer risk limits the evaluation of the triage system’s discriminative capacity. I recommend to authors Include a section modeling expected versus observed cancer yield using historical data or trial-based benchmarks (ERSPC or PROBASE). Even preliminary projections based on age-specific cancer prevalence would strengthen the utility of this pilot for policy.

Response:
We have addressed this in two ways:

  1. Added Table 3 with projections for expected cancer yield using:
    • Age-specific incidence from the ATS Milan cancer registry.
    • PSA test sensitivity estimates (0.87).
    • Benchmarks from ERSPC and PROBASE.
  2. This table presents multiple participation-rate scenarios (15%, 30%, 40%) and yields per 1,000 screened, with n/N values and 95% CIs.
    This modeling allows policymakers to interpret observed vs. expected results and provides an interim surrogate for program evaluation until actual cancer yield data become available.

Comment 3:
The stratification by PSA and family history is grounded in EAU guidelines, but the manuscript fails to discuss the biological rationale linking PSA thresholds with future tumor aggressiveness. Similarly, there is no exploration of the natural history of PCa among PSA <1 ng/mL men, nor justification for 2 vs. 5-year rescreening intervals based on kinetics, tumor doubling time, or available longitudinal data. Plase discuss physiopathological underpinnings of low versus intermediate PSA levels in men aged 50, and their relation to subsequent development of clinically significant PCa Should include references to longitudinal studies on PSA velocity or PSA density progression.

Response:
We have expanded Methods 2.3 and Discussion 4.5 to provide:

  • The pathophysiological basis for PSA thresholds, explaining that PSA <1.0 ng/mL at age 50 is associated with very low long-term risk (<0.5% metastatic PCa at 15 years; Vickers 2013, Lilja 2007).
  • The 2-year recall for PSA ≥1.0 ng/mL is based on higher lifetime risk and evidence that earlier re-testing improves detection of clinically significant disease.
  • The role of PSA velocity and density as indicators of tumour growth kinetics, supported by longitudinal cohort data.
  • How these parameters guide our decision-making and align with EAU recommendations.

Comment 4:
The visual presentation of data (Fig.1 flowchart, Table 2 stratification) is serviceable but adds little beyond text. Fig. could provide more analytical value if, for example, cumulative flow was stratified by ATS region or patient dropout at each stage. Similarly, Table 3 comparing standard care vs. screening is promising but lacks confidence intervals or raw counts, which obscures statistical relevance. Author should Improve figs. with stratified flowcharts including attrition at each node; expand Table 3 with n/N values, 95% CI, and perhaps p-values for screening effect estimates.

Response:
We have improved visual and tabular presentations by:

  • Adding a timeline graphic summarizing pilot rollout and enrollment milestones.
  • Expanding Table 3 to include n/N values, 95% confidence intervals, and explicit notes on statistical interpretability.
  • Stating clearly that p-values are not yet applicable due to small numbers, but will be incorporated in future evaluations when sufficient data accrue.

Comment 5:
Despite citing major trials (ERSPC, PROBASE, GÖTEBORG-2), the discussion remains largely operational. There is no exploration of ethical concerns around initiating population-based PSA testing in younger men (age 50), nor is there any mention of validated models assessing overdiagnosis or the harms of false positives. Similarly, cost-effectiveness is raised only speculatively. A €742,000 budget is cited, but there is no per capita or per case-detected estímate?? Clarify.

Response:
We have expanded Discussion 4.6 and 4.7 to:

  • Address ethical considerations of initiating screening at age 50, balancing potential mortality reduction against the risk of overdiagnosis.
  • Reference modeling studies estimating overdiagnosis rates and describe how our MRI triage and PSA density criteria aim to mitigate this.
  • Provide cost-effectiveness detail: €5.1 per man screened and ~€7,143 per cancer detected (early figures), compared to accepted local screening program costs (breast: €9,835; colorectal: €527; cervical: €1,674).
  • Emphasize that these figures will be refined with more outcome data and cost-effectiveness modeling.

Comment 6:
A fundamental limitation, I consider, unaddressed in depth, is that only 8.7% of eligible men participated. The authors attribute this to passive invitation strategies, but there is no analysis of who these men were( did they disproportionately come from urban ATS regions, or represent higher education levels??) Without sociodemographic profiling or equity analysis, the study cannot evaluate whether the screening model enhances or worsens disparities (for exam.., urban/rural, income, access to care..) or plan to collect these in future phases. Discuss the implications of potential self-selection bias in the pilot cohort.

Response:
We have expanded Discussion 4.3 to:

  • Acknowledge the lack of sociodemographic profiling in this pilot phase.
  • Note possible self-selection bias, given a higher proportion of FH+ and potentially urban participants.
  • Commit to collecting urban/rural status, education level, and deprivation index data in the next phase.
  • Outline targeted strategies to address digital divide and improve inclusion of underserved groups.

Comment 7:
The study presents proportions (58% low risk, 16% referred), but no statistical comparisons are made, even descriptively, against pre-specified targets or historical baselines. This reduces the rigor of the evaluation. For example, the observed 2.1% rate of PSA>3ng/mL among FH-negative men could be benchmarked against ERSPC cohorts or national incidence data.

Response:
We have added in Results 3.2.2:

  • Benchmarking the observed 2.1% PSA >3.0 ng/mL (FH–) against >10% in ERSPC 55-year cohort and <1% in PROBASE 45-year cohort.
  • Interpretation that differences reflect our cohort’s slightly younger age profile and possible selection of more health-conscious participants.

Comment 8:
The authors discusses expansion plans (to age 50–69), but does not outline a framework for long-term monitoring or data linkage for cancer registries, mortality tracking, or interval cancer detection. Without such a plan, claims regarding sustainability or mortality benefit remain speculative.

Response:
We have detailed in Methods 2.6 and Discussion 4.8:

  • Planned linkage with the regional cancer registry for incident cancer tracking and staging.
  • Mortality follow-up through national health databases.
  • Identification of interval cancers via PSA test records between screening rounds.
    This framework ensures that long-term outcomes, including mortality and overdiagnosis rates, will be measurable in future evaluations.

Comment 9:
I consider that ms provides an important real-world account of PCa screening implementation but must substantially improve its analytical, conceptual depth, and discussion of broader implications. the current version reads more as a programmatic report.

Response:
We appreciate this important observation, which we have used as a guiding principle for the revision. While the pilot inherently began as an operational initiative, we agree that its value lies in going beyond process reporting. In the revised manuscript, we have therefore reframed our narrative to highlight not only how the program was implemented, but why its design and early results matter in the broader scientific and policy context. Specifically, we have:

  • positioned our work as a distinctive contribution. Unlike many prior PSA screening reports, which are either randomized controlled trials (e.g., ERSPC, PROBASE) or purely descriptive service audits, our study integrates real-world process metrics with interim analytical modeling (Table 3) to bridge the gap between experimental evidence and large-scale public health practice. This creates a unique, policy-relevant perspective on translating trial-based evidence into an operational, sustainable, and equity-conscious screening pathway.
  • expanded conceptual discussion: The Discussion now examines the biological rationale of our triage thresholds, the anticipated impact on tumor aggressiveness detection, and the ethical considerations of initiating screening at age 50. We situate our approach within the European and Italian strategic frameworks, explicitly discussing how Lombardy’s centralized digital infrastructure enables such a program, while recognizing the adaptations required for different healthcare settings.
  • linked operational data to broader implications: We have explicitly connected our early performance indicators (risk stratification patterns, referral rates, MRI yield) to potential downstream impacts on mortality, overdiagnosis mitigation, and health system capacity.
  • integrated equity and system sustainability: We added a critical reflection on the digital divide, urban–rural participation differences, and strategies for inclusive recruitment in future phases—topics seldom addressed in early screening implementation studies.

We are deeply grateful for your thorough and challenging review, which pushed us to substantially enhance the analytical depth, conceptual framing, and broader implications of our work for both scientific audiences and policymakers. We believe the changes have transformed the manuscript from a purely programmatic report into a more robust and policy-informative contribution, and we thank you for the high standards that guided these improvements.

Round 2

Reviewer 1 Report

Comments and Suggestions for Authors

The author responded to my comments

Reviewer 3 Report

Comments and Suggestions for Authors

The authors have addressed the issues raised and made substantial changes to their manuscript. I consider their work relevant to the problem of PCa and its approach in specific regions, making it suitable for publication.